# Batch Normalization Biases Residual Blocks Towards the Identity Function in Deep Networks

**Soham De**
DeepMind, London
sohamde@google.com

**Samuel L. Smith**
DeepMind, London
slsmith@google.com

## Abstract

Batch normalization dramatically increases the largest trainable depth of residual networks, and this benefit has been crucial to the empirical success of deep residual networks on a wide range of benchmarks. We show that this key benefit arises because, at initialization, batch normalization downscales the residual branch relative to the skip connection, by a normalizing factor on the order of the square root of the network depth. This ensures that, early in training, the function computed by normalized residual blocks in deep networks is close to the identity function (on average). We use this insight to develop a simple initialization scheme that can train deep residual networks without normalization. We also provide a detailed empirical study of residual networks, which clarifies that, although batch normalized networks can be trained with larger learning rates, this effect is only beneficial in specific compute regimes, and has minimal benefits when the batch size is small.

## 1 Introduction

The combination of skip connections [1–3] and batch normalization [4] dramatically increases the largest trainable depth of neural networks. Although the origin of this effect is poorly understood, it has led to a rapid improvement in the performance of deep networks on popular benchmarks [5, 6]. Following the introduction of layer normalization [7] and the transformer architecture [8, 9], almost all state-of-the-art networks currently contain both skip connections and normalization layers.

**Our contributions.** This paper provides a simple explanation for why batch normalized deep residual networks are easily trainable. We prove that batch normalization downscales the hidden activations on the residual branch by a factor on the order of the square root of the network depth (at initialization). Therefore, as the depth of a residual network is increased, the residual blocks are increasingly dominated by the skip connection, which drives the functions computed by residual blocks closer to the identity, preserving signal propagation and ensuring well-behaved gradients [10–15].

If our theory is correct, it should be possible to train deep residual networks without normalization, simply by downscaling the residual branch. Therefore, to verify our analysis, we introduce a one-line code change ("SkipInit") which imposes this property at initialization, and we confirm that this alternative scheme can train one thousand layer deep residual networks without normalization.

In addition, we provide a detailed empirical study of residual networks at a wide range of batch sizes. This study demonstrates that, although batch normalization does enable us to train residual networks with larger learning rates, we only benefit from using large learning rates in practice if the batch size is also large. When the batch size is small, both normalized and unnormalized networks have similar optimal learning rates (which are typically much smaller than the largest stable learning rates) and yet normalized networks still achieve significantly higher test accuracies and lower training losses. These experiments demonstrate that, in residual networks, increasing the largest stable learning rate is not the primary benefit of batch normalization, contrary to the claims made in prior work [16, 17].

**Paper layout.** In section 2, we prove that residual blocks containing identity skip connections and normalization layers are biased towards the identity function in deep networks (at initialization). To confirm that this property explains why deep normalized residual networks are trainable, we propose a simple alternative to normalization ("SkipInit") that shares the same property at initialization, and we provide an empirical study of normalized residual networks and SkipInit at a range of network depths. In section 3, we study the performance of residual networks at a range of batch sizes, in order to clarify when normalized networks benefit from large learning rates. We study the regularization benefits of batch normalization in section 4 and we compare the performance of batch normalization, SkipInit and Fixup [18] on ImageNet in section 5. We discuss related work in section 6.

## 2 Why are deep normalized residual networks trainable?

### 2.1 Theoretical analysis at initialization

Residual networks (ResNets) [2, 3] contain a sequence of residual blocks, which are composed of a "residual branch" comprising a number of convolutions, normalization layers and non-linearities, as well as a "skip connection", which is usually just the identity (See figure 1). While introducing skip connections shortens the effective depth of the network, on their own they only increase the trainable depth by roughly a factor of two [15]. Normalized residual networks, on the other hand, can be trained for depths significantly deeper than twice the depth of their non-residual counterparts [3, 18].

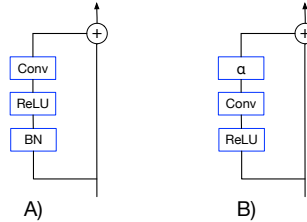

Figure 1: A) A residual block with batch normalization. It is common practice to include two convolutions on the residual branch; we show one convolution for simplicity. B) SkipInit replaces batch normalization by a single learnable scalar $\alpha$. We set $\alpha = 0$ (or a small constant) at initialization.

To understand this effect, we analyze the variance of hidden activations at initialization. For clarity, we focus here on the variance of a single training example, but we discuss the variance across batches of training examples (which share the same random weights) in appendix C. Let $x_i^\ell$ denote the $i$-th component of the input to the $\ell$-th residual block, where $x^1$ denotes the input to the model with $\mathbf{E}(x_i^1) = 0$ and $\mathbf{Var}(x_i^1) = 1$ for each independent component $i$. Let $f^\ell$ denote the function computed by the residual branch of the $\ell$-th residual block, $x_i^+ = \max(x_i, 0)$ denote the output of the ReLU, and $\mathcal{B}$ denote the batch normalization operation (for completeness, we define batch normalization formally in appendix A). For simplicity, we assume that there is a single linear layer on each residual branch, such that for normalized networks, $f^\ell(x^\ell) = W^\ell \mathcal{B}(x^\ell)^+$, and for unnormalized networks $f^\ell(x^\ell) = W^\ell x^{\ell+}$. We also assume that each component of $W^\ell$ is independently sampled from $\mathcal{N}(0, 2/\text{fan-in})$ (He Initialization) [19].[1] Thus, given $x^\ell$, the mean of the $i$-th coordinate of the output of a residual branch $\mathbf{E}(f_i^\ell(x^\ell)|x^\ell) = 0$. Since $x^{\ell+1} = x^\ell + f^\ell(x^\ell)$, this implies $\mathbf{E}(x_i^\ell) = 0$ for all $\ell$. The covariance between the residual branch and the skip connection $\mathbf{Cov}(f_i^\ell(x^\ell), x_i^\ell) = 0$, and thus the variance of the hidden activations, $\mathbf{Var}(x_i^{\ell+1}) = \mathbf{Var}(x_i^\ell) + \mathbf{Var}(f_i^\ell(x^\ell))$. We conclude:

**Unnormalized networks:** If the residual branch is unnormalized, the variance of the residual branch, $\mathbf{Var}(f_i^\ell(x^\ell)) = \sum_j^{\text{fan-in}} \mathbf{Var}(W_{ij}^\ell) \cdot \mathbf{E}((x_j^{\ell+})^2) = 2 \cdot \mathbf{E}((x_i^{\ell+})^2) = \mathbf{Var}(x_i^\ell)$. This has two implications. First, the variance of the hidden activations explode exponentially with depth, $\mathbf{Var}(x_i^{\ell+1}) = 2 \cdot \mathbf{Var}(x_i^\ell) = 2^\ell$. One can prevent this explosion by introducing a factor of $(1/\sqrt{2})$ at the end of each residual block, such that $x^{\ell+1} = (x^\ell + f^\ell(x^\ell))/\sqrt{2}$. Second, since $\mathbf{Var}(f_i^\ell(x^\ell)) = \mathbf{Var}(x_i^\ell)$, the residual branch and the skip connection contribute equally to the output of the residual block. This ensures that the function computed by the residual block is far from the identity function.

**Normalized networks:** If the residual branch is normalized, the variance of the output of the residual branch $\mathbf{Var}(f_i^\ell(x^\ell)) = \sum_j^{\text{fan-in}} \mathbf{Var}(W_{ij}^\ell) \cdot \mathbf{E}((\mathcal{B}(x^\ell)_j^+)^2) = \mathbf{Var}(\mathcal{B}(x^\ell)_i) \approx 1$.[2] Thus, the variance of the input to the $\ell$-th residual block, $\mathbf{Var}(x_i^\ell) \approx \mathbf{Var}(x_i^{\ell-1}) + 1$, which implies $\mathbf{Var}(x_i^\ell) \approx \ell$.

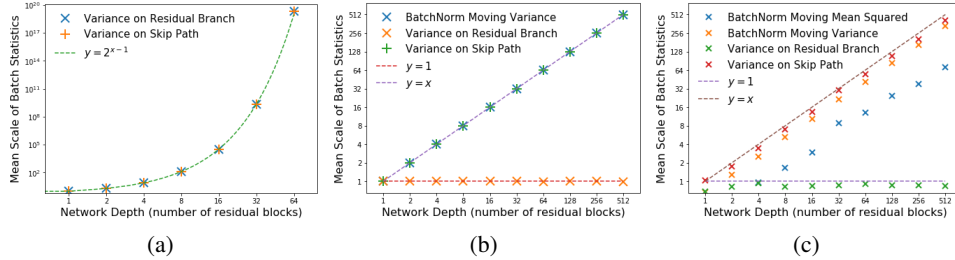

Figure 2: We empirically evaluate the dependence of the variance of the hidden activations on the depth of the residual block at initialization (See appendix B for details). In (a), we consider a fully connected ResNet with linear activations without any normalization, evaluated on random Gaussian inputs. In (b), we consider the same ResNet but with one normalization layer on each residual branch. The squared BatchNorm moving mean is close to zero (not shown). In (c), we consider a batch normalized convolutional residual network with ReLU activations, evaluated on CIFAR-10.

Surprisingly, the growth in the variance of the hidden activations is beneficial, because if $\mathbf{Var}(x_i^\ell) \approx \ell$, then the batch normalization operation $\mathcal{B}$ must suppress the variance of the $\ell$-th residual branch by a factor of $\ell$ (hidden activations are suppressed by $\sqrt{\ell}$). Consequently, the residual branch contributes only a $1/(\ell+1)$ fraction of the variance in the output of the $\ell$-th residual block. This ensures that, at initialization, the outputs of most residual blocks in a deep normalized ResNet are dominated by the skip connection, which biases the function computed by the residual block towards the identity.

The depth of a typical residual block is proportional to the total number of residual blocks $d$, which implies that batch normalization downscales residual branches by a factor on the order of $\sqrt{d}$. Although this is weaker than the factor of $d$ proposed in [12], we find empirically in section 2.3 that it is sufficiently strong to train deep residual networks with 1000 layers. We emphasize that while our analysis only explicitly considers the propagation of the signal on the forward pass, residual blocks dominated by the skip path on the forward pass will also preserve signal propagation on the backward pass. This is because, when the forward signal on the $\ell$-th residual branch is downscaled by a factor $\alpha$, the backward propagated signal through that branch will also be downscaled by a factor $\alpha$ [20].

To verify our analysis, we evaluate the variance of the hidden activations, as well as the batch normalization statistics, of three residual networks at initialization in figure 2. We define the networks in appendix B. In figure 2(a), we consider a fully connected linear unnormalized residual network, where we find that the variance on the skip path of the $\ell$-th residual block matches the variance of the residual branch and is equal to $2^{\ell-1}$, as predicted by our analysis. In figure 2(b), we consider a fully connected linear normalized residual network, where we find that the variance on the skip path of the $\ell$-th residual block is approximately equal to $\ell$, while the variance at the end of each residual branch is approximately 1. The batch normalization moving variance on the $\ell$-th residual block is also approximately equal to $\ell$, confirming that batch normalization downscales the residual branch by a factor of $\sqrt{\ell}$ as predicted. In figure 2(c), we consider a normalized convolutional residual network with ReLU activations evaluated on CIFAR-10. The variance on the skip path remains proportional to the depth of the residual block, with a coefficient slightly below 1 (likely due to zero padding at the image boundary). The batch normalization moving variance is also proportional to depth, but slightly smaller than the variance across channels on the skip path. We show in appendix C that this occurs because ReLU activations introduce correlations between different examples in the mini-batch. These correlations also cause the square of the batch normalization moving mean to grow with depth.

## 2.2 SkipInit; an initialization scheme to verify our analysis

We claim above that batch normalization enables us to train deep residual networks, because (in expectation) it downscales the residual branch at initialization by a normalizing factor on the order of the square root of the network depth. To provide further evidence for this claim, we now propose a simple initialization scheme that can train deep residual networks without normalization, "SkipInit":

*SkipInit: Include a learnable scalar multiplier at the end of each residual branch, initialized to $\alpha$.*

After normalization is removed, it should be possible to implement SkipInit as a one line code change. In section 2.3, we show that we can train deep residual networks, so long as $\alpha$ is initialized at a value

Table 1: Batch normalization enables us to train deep residual networks. We can recover this benefit without normalization if we introduce a scalar multiplier $\alpha$ on the end of the residual branch and initialize $\alpha = (1/\sqrt{d})$ or smaller (where $d$ is the number of residual blocks). In practice, we advocate initializing $\alpha = 0$. We provide optimal test accuracies and optimal learning rates with error bars. Note that we do not provide results in cases where the test accuracy was frozen at random initialization throughout training for all learning rates in the range $2^{-10}$ to $2^2$ (i.e., in cases where training failed).

**Batch Normalization**

| Depth | Test accuracy | Learning rate |
|---|---|---|
| 16 | $93.5 \pm 0.1$ | $2^{-1}$ ($2^{-1}$ to $2^{-1}$) |
| 100 | $94.7 \pm 0.1$ | $2^{-1}$ ($2^{-2}$ to $2^{-0}$) |
| 1000 | $94.6 \pm 0.1$ | $2^{-2}$ ($2^{-3}$ to $2^{-0}$) |

**SkipInit ($\alpha = 1$)**

| Depth | Test accuracy | Learning rate |
|---|---|---|
| 16 | $93.0 \pm 0.1$ | $2^{-2}$ ($2^{-2}$ to $2^{-1}$) |
| 100 | $-$ | $-$ |
| 1000 | $-$ | $-$ |

**SkipInit ($\alpha = 1/\sqrt{d}$)**

| Depth | Test accuracy | Learning rate |
|---|---|---|
| 16 | $93.0 \pm 0.1$ | $2^{-2}$ ($2^{-2}$ to $2^{-1}$) |
| 100 | $94.2 \pm 0.1$ | $2^{-1}$ ($2^{-2}$ to $2^{-1}$) |
| 1000 | $94.2 \pm 0.0$ | $2^{-1}$ ($2^{-2}$ to $2^{-1}$) |

**Divide residual block by $\sqrt{2}$**

| Depth | Test accuracy | Learning rate |
|---|---|---|
| 16 | $92.4 \pm 0.1$ | $2^{-2}$ ($2^{-2}$ to $2^{-1}$) |
| 100 | $88.9 \pm 0.4$ | $2^{-5}$ ($2^{-5}$ to $2^{-5}$) |
| 1000 | $-$ | $-$ |

**SkipInit ($\alpha = 0$)**

| Depth | Test accuracy | Learning rate |
|---|---|---|
| 16 | $93.3 \pm 0.1$ | $2^{-2}$ ($2^{-2}$ to $2^{-2}$) |
| 100 | $94.2 \pm 0.1$ | $2^{-2}$ ($2^{-2}$ to $2^{-2}$) |
| 1000 | $94.3 \pm 0.2$ | $2^{-2}$ ($2^{-3}$ to $2^{-1}$) |

**SkipInit without L2 ($\alpha = 0$)**

| Depth | Test accuracy | Learning rate |
|---|---|---|
| 16 | $89.8 \pm 0.2$ | $2^{-3}$ ($2^{-3}$ to $2^{-3}$) |
| 100 | $91.7 \pm 0.2$ | $2^{-2}$ ($2^{-2}$ to $2^{-2}$) |
| 1000 | $92.1 \pm 0.1$ | $2^{-2}$ ($2^{-2}$ to $2^{-2}$) |

of $(1/\sqrt{d})$ or smaller, where $d$ denotes the total number of residual blocks (see table 1). Notice that this observation agrees exactly with our analysis of deep normalized residual networks in section 2.1. In practice, we recommend setting $\alpha = 0$, so that the residual block represents the identity function at initialization. This choice is also simpler to apply, since it ensures the initialization scheme is independent of network depth. We note that SkipInit is designed for residual networks that contain an identity skip connection such as the ResNet-V2 [3] or Wide-ResNet architectures [21]. We discuss how to extend SkipInit to the original ResNet-V1 [2] formulation of residual networks in appendix F.

## 2.3 An empirical study of residual networks at a wide range of network depths

We empirically verify the claims made above by studying the minimal components required to train deep residual networks. In table 1, we report the mean test accuracy of an $n$-2 Wide-ResNet [21], trained on CIFAR-10 for 200 epochs at batch size 64 at a range of depths $n$ between 16 and 1000 layers. At each depth, we train the network 7 times for a range of learning rates on a logarithmic grid, and we measure the mean and standard deviation of the test accuracy for the best 5 runs (this procedure ensures that our results are not corrupted by outliers or failed runs). The optimal test accuracy is the mean performance at the learning rate whose mean test accuracy was highest, and we always verify that the optimal learning rates are not at the boundary of our grid search. Here and throughout this paper, we use SGD with heavy ball momentum, and fix the momentum coefficient $m = 0.9$. Although we tune the learning rate on the test set, we emphasize that our goal is not to achieve state of the art results. Our goal is to compare the performance of different training procedures, and we apply the same experimental protocol in each case. We hold the learning rate constant for 100 epochs, before dropping the learning rate by a factor of 2 every 10 epochs. This simple schedule achieves higher test accuracies than the original 3 drops schedule proposed in [2]. We apply data augmentation including per-image standardization, padding, random crops and left-right flips. We use L2 regularization with a coefficient of $5 \times 10^{-4}$, and we initialize convolutional layers using He initialization [19]. We provide the corresponding optimal training losses in appendix D.

As expected, batch normalized Wide-ResNets are trainable for a wide range of depths, and the optimal learning rate is only weakly dependent on the depth. We can recover this effect without normalization by incorporating SkipInit and initializing $\alpha = (1/\sqrt{d})$ or smaller, where $d$ denotes the number of residual blocks. This provides strong evidence to support our claim that batch normalization enables us to train deep residual networks by biasing residual blocks towards the skip path at initialization.

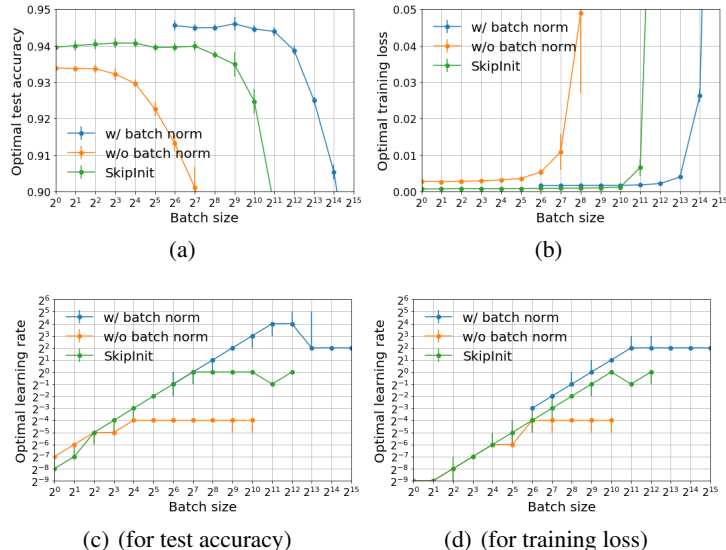

<div align="center">(a)                              (b)</div>

<div align="center">(c) (for test accuracy)         (d) (for training loss)</div>

Figure 3: In (a), we achieve higher test accuracies with batch normalization than without batch normalization, and we are also able to train efficiently at much larger batch sizes. SkipInit substantially reduces the gap in performance for small/moderate batch sizes, but it still under-performs batch normalization when the batch size is large. In (b), SkipInit achieves smaller training losses than batch normalization for batch sizes $b \lesssim 1024$. We provide the test accuracy at the learning rate for which the test accuracy was maximized, and the training loss at the learning rate for which the training loss was minimized. To help interpret these results, we also provide the optimal learning rates in figures (c) and (d). When the batch size is small, all three methods have similar optimal learning rates (which are much smaller than the maximum stable learning rate for each method), but batch normalization and SkipInit are able to scale to larger learning rates when the batch size is large.

Just like normalized networks, the optimal learning rate with SkipInit is almost independent of the network depth. SkipInit slightly under-performs batch normalization on the test set at all depths, although we show in appendix D that it achieves similar training losses to normalized networks.

For completeness, we verify in table 1 that one cannot train deep residual networks with SkipInit if $\alpha = 1$. We also show that for unnormalized residual networks, it is not sufficient merely to ensure the activations do not explode on the forward pass (which can be achieved by multiplying the output of each residual block by $(1/\sqrt{2})$). This confirms that ensuring stable forward propagation of the signal is not sufficient for trainability. Additionally, we noticed that, at initialization, the loss in deep networks is dominated by the L2 regularization term, causing the weights to shrink rapidly early in training. To clarify whether this effect is necessary, we evaluated SkipInit ($\alpha = 0$) without L2 regularization, and find that L2 regularization is not necessary for trainability. This demonstrates that we can train deep residual networks without normalization and without reducing the scale of the weights at initialization, solely by downscaling the hidden activations on the residual branch. To further test the theory that downscaling the residual branch is the key benefit of batch normalization in deep ResNets, we tried several other variations of batch-normalized ResNets, which we present in appendix D. We find that variants of batch-normalized ResNets which do not downscale the residual branch relative to the skip path are not trainable for large depths (e.g. networks that place normalization layers on the skip path). We provide additional results on CIFAR-100 in appendix E.

## 3   When can normalized networks benefit from large learning rates?

In two widely read papers, Santurkar et al. [16] and Bjorck et al. [17] argued that the primary benefit of batch normalization is that it improves the conditioning of the loss landscape, which allows us to train stably with larger learning rates. However, this claim seems incompatible with a number of recent papers studying optimization in deep learning [22–29]. These papers argue that if we train for a fixed number of epochs (as is common in practice), then when the batch size is small, the optimal

learning rate is significantly smaller than the largest stable learning rate, since it is constrained by the noise in the gradient estimate. In this small batch regime, the optimal learning rate is usually proportional to the batch size [29–31]. Meanwhile the conditioning of the loss sets the maximum stable learning rate [26–29], and this controls how large we can make the batch size before the performance of the model begins to degrade under a fixed epoch budget. If this perspective is correct, we would expect large stable learning rates to be beneficial only when the batch size is also large. In this section, we clarify the role of large learning rates in normalized networks by studying residual networks with and without batch normalization at a wide range of batch sizes.

In figure 3, we provide results for a 16-4 Wide-ResNet, trained on CIFAR-10 for 200 epochs at a wide range of batch sizes and learning rates. We follow the same experimental protocol described in section 2.3, however we average over the best 12 out of 15 runs. To enable us to consider extremely large batch sizes on a single GPU, we evaluate the batch statistics over a "ghost batch size" of 64, before accumulating gradients to form larger batches, as is standard practice [32]. We therefore are unable to consider batch sizes below 64 with batch normalization. Note that we repeat this experiment in the small batch limit in section 4, where we evaluate the batch statistics over the full training batch.

Unsurprisingly, the performance with batch normalization is better than the performance without batch normalization on both the test set and the training set at all batch sizes.[3] However, both with and without batch normalization, the optimal test accuracy is independent of batch size in the small batch limit, before beginning to decrease when the batch size exceeds some critical threshold.[4] Crucially, this threshold is significantly larger when batch normalization is used, which demonstrates that one can efficiently scale training to larger batch sizes in normalized networks. SkipInit reduces the gap in test accuracy between normalized and unnormalized networks, and it achieves smaller training losses than batch normalization when the batch size is small ($b \lesssim 1024$). However similar to unnormalized networks, it still performs worse than normalized networks when the batch size is very large.

To explain why normalized networks can scale training to larger batch sizes, we provide the optimal learning rates that maximize the test accuracy and minimize the training loss in figures 3(c) and 3(d). When the batch size is small, the optimal learning rates for all three methods are proportional to the batch size and are similar to each other. Crucially, the optimal learning rates are much smaller than the largest stable learning rate for each method. On the other hand, when the batch size is large, the optimal learning rates are independent of batch size [26, 27], and normalized networks use larger learning rates. Intuitively, this transition occurs when we reach the maximum stable learning rate, above which training diverges [28]. Our results confirm that batch normalized networks have a larger maximum stable learning rate than SkipInit networks, which have a larger maximum stable learning rate than unnormalized networks. This explains why batch normalized networks were able to efficiently scale training to larger batch sizes. Crucially however, our experiments confirm that batch normalized networks do not benefit from the use of large learning rates when the batch size is small.

Furthermore, under a fixed epoch budget, the highest test accuracies for all three methods are always achieved in the small batch limit with small learning rates, and the test accuracy never increases when the batch size rises. We therefore conclude that large learning rates are not the primary benefit of batch normalization in residual networks, contradicting the claims of earlier work [16, 17]. The primary benefit of batch normalization is that it biases the residual blocks in deep residual networks towards the identity function, thus enabling us to train significantly deeper networks. To emphasize this claim, we show in the next section that the gap in test accuracy between batch normalization and SkipInit in the small batch limit can be further reduced with additional regularization. We provide additional results sweeping the batch size on a 28-10 Wide-ResNet on CIFAR-100 in appendix E.

## 4    On the regularization benefit of batch normalization

It is widely known that batch normalization can have a regularizing effect [32]. Most authors believe that this benefit arises from the noise that arises when the batch statistics are estimated on a subset of the full training set [33]. In this section, we study this regularization benefit at a range of batch sizes. Unlike the previous section (which used a "ghost batch size" of 64 [32]), in this section we will

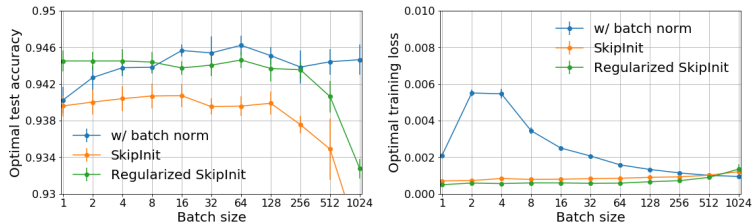

Figure 4: To study the regularization benefits of batch normalization, we evaluate the batch statistics over the full batch, allowing us to consider any batch size $b \geq 1$. The training loss falls as the batch size increases, but the test accuracy is maximized for an intermediate batch size, $b \approx 64$. Regularized SkipInit outperforms batch normalization on the test set for small batch sizes.

evaluate the batch statistics of normalized networks over the entire mini-batch. We introduced SkipInit in section 2.2, which ensures that very deep unnormalized ResNets are trainable. To attempt to recover the additional regularization benefits of batch normalization, we now introduce "Regularized SkipInit". This scheme includes SkipInit ($\alpha = 0$), but also introduces biases to all convolutions and applies a single Dropout layer [34] before the softmax (We use drop probability 0.6 in this section).

In figure 4, we provide the performance of our 16-4 Wide-ResNet at a range of batch sizes in the small batch limit (note that batch normalization reduces to instance normalization when the batch size $b = 1$). We provide the corresponding optimal learning rates in appendix D. The test accuracy of batch normalized networks initially improves as the batch size rises, before decaying for batch sizes $b \gtrsim 64$. Meanwhile, the training loss increases as the batch size rises from 1 to 2, but then decreases consistently as the batch size rises further. This confirms that the uncertainty in the estimate of the batch statistics does have a generalization benefit if properly tuned (This is also why we chose a ghost batch size of 64 in section 3). The performance of SkipInit and Regularized SkipInit are independent of batch size in the small batch limit, and Regularized SkipInit achieves higher test accuracies than batch normalization when the batch size is very small. Note that we introduced Dropout [34] to show that extra regularization may be necessary to close the performance gap between normalized and SkipInit networks, but more sophisticated regularizers would likely achieve higher test accuracies. We provide additional results studying this regularization effect on CIFAR-100 in appendix E.

## 5    A comparison on ImageNet

In this section, we compare the performance of batch normalization and SkipInit on ImageNet. For completeness, we also compare to the recently proposed Fixup initialization [18]. Since SkipInit is designed for residual networks with an identity skip connection, we consider the ResNet50-V2 architecture [3]. We provide additional experiments on ResNet50-V1 [2] in appendix F. We use the original architectures and match the performance reported by [35] (we do not apply the popular modifications to these architectures described in [22]). We train for 90 epochs, and when batch normalization is used we set the ghost batch size to 256. The learning rate is linearly increased from 0 to the specified value over the first 5 epochs of training [22], and then held constant for 40 epochs, before decaying it by a factor of 2 every 5 epochs. As before, we tune the learning rate at each batch size on a logarithmic grid. We provide the optimal validation accuracies in table 2. We found that adding biases to the convolutional layers led to a small boost in accuracy for SkipInit, and we therefore included biases in all SkipInit runs. SkipInit and Fixup match the performance of batch normalization at the standard batch size of 256, however both SkipInit and Fixup perform worse than batch normalization when the batch size is very large. Both SkipInit and Fixup achieve higher test accuracies than batch normalization with extra regularization (Dropout) for small batch sizes. We include code for our Tensorflow [36] implementation of ResNet50-V2 with SkipInit in appendix G.

## 6    Related work

In recent years, almost all state-of-the-art models have involved applying some kind of normalization scheme [4, 7, 37–39] in combination with skip connections [1–3, 8, 9]. Although some authors have succeeded in training very deep networks without normalization layers or skip connections [14, 40],

Table 2: When training ResNet50-V2 on ImageNet, SkipInit and Fixup are competitive with batch normalization for small batch sizes, while batch normalization performs best when the batch size is large. SkipInit and Fixup both achieve higher validation accuracies than batch normalization with extra regularization. We train for 90 epochs and perform a grid search to identify the optimal learning rate which maximizes the top-1 validation accuracy. We perform a single run at each learning rate and report top-1 and top-5 accuracy scores. We use a drop probability of $0.2$ when Dropout is used.

| Test accuracy: | Batch size | | |
|---|---|---|---|
| | 256 | 1024 | 4096 |
| Batch normalization | 75.0 / 92.2 | 74.9 / 92.1 | 74.9 / 91.9 |
| Fixup | 74.8 / 91.8 | 74.6 / 91.7 | 73.0 / 90.6 |
| SkipInit + Biases | 74.9 / 91.9 | 74.6 / 91.8 | 70.8 / 89.2 |
| Fixup + Dropout | 75.8 / 92.5 | 75.6 / 92.5 | 74.8 / 91.8 |
| Regularized SkipInit | 75.6 / 92.4 | 75.5 / 92.5 | 72.7 / 90.7 |

these papers required careful orthogonal initialization schemes that are not compatible with ReLU activation functions. Balduzzi et al. [11] and Yang et al. [13] argued that ResNets with identity skip connections and batch normalization layers on the residual branch preserve correlations between different minibatches in deep networks, and Balduzzi et al. [11] suggested that this effect can be mimicked by initializing deep networks close to linear functions. However, even deep linear networks are difficult to train with Gaussian weights [12, 15, 40], which suggests that imposing linearity at initialization is not sufficient. Veit et al. [10] observed empirically that normalized residual networks are typically dominated by short paths, however they did not identify the cause of this effect. Some authors have studied initialization schemes which multiply the output of the residual branch by a fixed scalar (smaller than 1), without establishing a link to normalization methods [11, 12, 41–44].

Santurkar et al. [16] and Bjorck et al. [17] argued that batch normalization improves the conditioning of the loss landscape, which enables us to train with larger learning rates and converge in fewer parameter updates. Arora et al. [45] argued that batch normalization reduces the importance of tuning the learning rate, while Li and Arora [46] showed that models trained using batch normalization can converge even if the learning rate increases exponentially during training. A similar analysis also appears in [47], while Luo et al. [33] analyzed the regularization benefits of batch normalization.

Zhang et al. [18] proposed Fixup initialization, and confirmed that it can train both deep residual networks and deep transformers without normalization layers. Fixup contains four components:

1. The classification layer and final convolution of each residual branch are initialized to zero.
2. The initial weights of the remaining convolutions are scaled down by $d^{-1/(2m-2)}$, where $d$ denotes the number of residual branches and $m$ is the number of convolutions per branch.
3. A scalar multiplier is introduced at the end of each residual branch, intialized to one.
4. Scalar biases are introduced before every layer in the network, initialized to zero.

The authors do not relate these components to the influence of the batch normalization layers on the residual branch, or seek to explain why deep normalized ResNets are trainable. They argue that the second component of Fixup is essential, however our experiments in section 2.3 demonstrate that this component is not necessary to train deep residual networks at typical batch sizes. In practice, we have found that either component 1 or component 2 of Fixup on its own is sufficient in ResNet-V2 networks, since both components downscale the hidden activations on the residual branch (fulfilling the same role as SkipInit). We found in section 5 that SkipInit and Fixup have similar performance for small batch sizes but that Fixup slightly outperforms SkipInit when the batch size is large.

# 7    Discussion

Our work demonstrates that batch normalization has three main benefits. In order of importance,

1. Batch normalization can train deep residual networks (section 2).
2. Batch normalization increases the maximum stable learning rate (section 3).
3. Batch normalization has a regularizing effect (section 4).

This work explains benefit 1, by observing that batch normalization biases residual blocks towards the identity function at initialization. This ensures that deep residual networks have well-behaved gradients, enabling efficient training [10]–[15]. Furthermore, our argument naturally extends to other normalization variants and model architectures, including layer normalization [7] and "pre-norm" transformers [9] (where the normalization layers are on the residual branch). A single normalization layer per residual branch is sufficient, and normalization layers should not be placed on the skip path (as in the original transformer [8]). We can recover benefit 1 without normalization by introducing a learnable scalar multiplier on the residual branch initialized to zero. This simple change can train deep ResNets without normalization, and often enhances the performance of shallow ResNets.

The conditioning benefit (benefit 2) is not necessary when one trains with small batch sizes, but it remains beneficial when one wishes to train with large batch sizes. Since large batch sizes can be computed in parallel across multiple devices [22], this could make normalization necessary in time-critical situations, for instance if a production model is retrained frequently in response to changing user preferences. Also, since batch normalization has a regularizing effect (benefit 3), it may be necessary in some architectures if one wishes to achieve the highest possible test accuracy. Note however that one can sometimes exceed the test accuracy of normalized networks by introducing alternate regularizers (see section 5 or [18]). We therefore believe future work should focus on identifying an alternative to batch normalization that recovers its conditioning benefits.

We would like to comment briefly on the similarity between SkipInit for residual networks, and Orthogonal initialization of vanilla fully connected tanh networks [40]. Orthogonal initialization is currently the only initialization scheme capable of training deep networks without skip connections. It initializes the weights of each layer as an orthogonal matrix, such that the activations after a linear layer are a rotation (or reflection) of the activations before the layer. Meanwhile, the tanh non-linearity is approximately equal to the identity for small activations over a region of scale 1 around the origin. Intuitively, if the incoming activations are mean centered with scale 1, they will pass through the non-linearity almost unchanged. Since rotations compose, the approximate action of the entire network at initialization is to rotate (or reflect) the input. Like residual blocks with SkipInit, the influence of a fully connected layer with orthogonal weights will therefore be close to the identity in function space. However ReLUs are not compatible with orthogonal initialization, since they are not linear about the origin, which has limited the use of orthogonal initialization in practice.

**To conclude.** Batch normalization biases the residual blocks of deep residual networks towards the identity function (at initialization). This ensures that the network has well behaved-gradients, and it is therefore a major factor behind the excellent empirical performance of normalized residual networks in practice. We show that one can recover this benefit in unnormalized residual networks with a one line code change to the architecture ("SkipInit"). In addition, we clarify that, although batch normalized networks can be trained with larger learning rates than unnormalized networks, this is only useful for large batch sizes and does not have practical benefits when the batch size is small.

## Broader impact

This work seeks to develop fundamental understanding by identifying the benefits batch normalization brings when training residual networks. We do not foresee any specific negative consequences of this work, although we hope that fundamental understanding may help drive future progress in the field.

## Funding disclosure

All authors are employees of DeepMind, which was the sole source of funding for this work. No authors have any competing interests.

## Acknowledgements

We thank Jeff Donahue, Chris Maddison, Erich Elsen, James Martens, Razvan Pascanu, Chongli Qin, Karen Simonyan, Yann Dauphin, Esme Sutherland and Yee Whye Teh for various discussions that have helped improve the paper.

## Footnotes

[1]fan-in denotes the number of incoming network connections to the layer.

[2]The approximation is tight when the batch size for computing the batch statistics is large.

[3]Note that we plot the training loss excluding the L2 regularization term in figure 3. Normalized networks often achieve smaller L2 losses because the network function is independent of the scale of the weights.

[4]As the batch size grows, the number of parameter updates decreases since the number of training epochs is fixed. We note that the performance might not degrade with batch size under a constant step budget [25].

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
