[Supplementary Material]

## A  Definition of a batch normalization layer

When applying batch normalization to convolutional layers, the inputs and outputs of normalization layers are 4-dimensional tensors, which we denote by $I_{b,x,y,c}$ and $O_{b,x,y,c}$. Here $b$ denotes the batch dimension, $c$ denotes the channels, and $x$ and $y$ are the two spatial dimensions. Batch normalization [4] applies the same normalization to every input in the same channel, such that:

$$O_{b,x,y,c} = \gamma_c \frac{I_{b,x,y,c} - \mu_c}{\sqrt{\sigma_c^2 + \epsilon}} + \beta_c.$$

Here, $\mu_c = \frac{1}{Z} \sum_{b,x,y} I_{b,x,y,c}$ denotes the per-channel mean, and $\sigma_c^2 = \frac{1}{Z} \sum_{b,x,y} I_{b,x,y,c}^2 - \mu_c^2$ denotes the per-channel variance, and $Z$ is the normalization constant summed over the minibatch $b$ and spatial dimensions $x$ and $y$. A small constant $\epsilon$ is included in the denominator for numerical stability. The "scale" and "shift" parameters, $\gamma_c$ and $\beta_c$, are learned during training. Typically, $\gamma_c$ is initialized to 1 and $\beta_c$ is initialized to 0, which is also what we consider in our analysis. Running averages of $\mu_c$ and $\sigma_c^2$ are also maintained during training, and these averages are used at test time to ensure the predictions are independent of other examples in the batch. For distributed training, the batch statistics are usually estimated locally on a subset of the training minibatch ("ghost batch normalization" [32]).

## B  Details of the residual networks used for figure 2

In figure 2 of the main text, we studied the variance of hidden activations and the batch statistics of residual blocks at a range of depths in three different architectures; a deep linear fully connected unnormalized residual network, a deep linear fully connected normalized residual network and a deep convolutional normalized residual network with ReLUs. We now define the three models in full.

**Deep fully connected linear residual network without normalization:** The inputs are 100 dimensional vectors composed of independent random samples from the unit normal distribution, and the batch size is 1000. These inputs first pass through a single fully connected linear layer of width 1000. We then apply a series of residual blocks. Each block contains an identity skip path, and a residual branch composed of a fully connected linear layer of width 1000. All linear layers are initialized with LeCun normal initialization [48] to preserve the variance in the absence of non-linearities.

**Deep fully connected linear residual network with batch normalization:** The inputs are 100 dimensional vectors composed of independent random samples from the unit normal distribution, and the batch size is 1000. These inputs first pass through a batch normalization layer and a single fully connected linear layer of width 1000. We then apply a series of residual blocks. Each block contains an identity skip path, and a residual branch composed of a batch normalization layer and a fully connected linear layer of width 1000. All linear layers are initialized with LeCun normal initialization [48] to preserve the variance in the absence of non-linearities.

**Deep convolutional ReLU residual network:** The inputs are batches of 100 images from the CIFAR-10 training set. We first apply a convolution of width 100 and stride 2, followed by a batch normalization layer, a ReLU non-linearity, and an additional convolution of width 100 and stride 2. We then apply a series of residual blocks. Each block contains an identity skip path, and a residual branch composed of a batch normalization layer, a ReLU non-linearity, and a convolution of width 100 and stride 1. All convolutions are initialized with He initialization [19].

In all three networks, we evaluate the variance at initialization on the skip path and at the end of the residual branch (we measure the empirical variance across multiple channels and multiple examples but for a single set of weights). For the two normalized networks, we also evaluate the mean moving variance and mean squared moving mean of the batch normalization layer (i.e., the mean value of the moving variance parameter and the mean value of the square of the moving mean, averaged over channels for a single set of weights). To obtain the batch normalization statistics, we set the momentum parameter of the batch normalization layers to 0, and then update the batch statistics once.

## C  The influence of ReLU non-linearities on batch normalization statistics

In the main text, we found that for the deep linear normalized residual network (figure 2(b)), the variance on the skip path is equal to the mean moving variance of the batch normalization layer, while

Figure 5: The batch statistics at initialization of a normalized deep fully connected network with ReLU non-linearities, evaluated on random inputs drawn from a Gaussian distribution.

the mean squared moving mean of the batch normalization layer is close to zero. However when we introduce ReLU non-linearities in the deep normalized convolutional residual network (figure 2(c)), the mean moving variance of the batch normalization layer is smaller than the variance across channels on the skip path, and the mean squared moving mean of the normalization layer grows proportional to the depth. To clarify the origin of this effect, we consider an additional fully connected deep normalized residual network with ReLU non-linearities. We form this network from the fully connected normalized linear residual network in appendix B by inserting a ReLU non-linearity after each normalization layer, and we replace LeCun initialization with He initialization. This network is easier to analyze than the convolutional network, but similar conclusions hold in both cases.

We provide the variance of the hidden activations and the batch statistics of this network in figure 5. The variance on the skip path in this network is approximately equal to the depth of the residual block $d$, while the variance at the end of the residual branch is approximately 1. This matches exactly our theoretical predictions in section 2 of the main text. Notice however that the mean moving variance of the batch normalization layer is approximately equal to $d(1 - 1/\pi)$, while the mean squared moving mean of the normalization layer is approximately equal to $d/\pi$. To understand these observations, we note that the outputs of a ReLU non-linearity have non-zero mean, and therefore the ReLU layer will cause the hidden activations of different examples on the same channel to become correlated (if the weights are fixed). Because of this, the variance across multiple examples and multiple channels becomes different from the variance across multiple examples for a single fixed channel.

To better understand this, we analyze this fully connected normalized ReLU residual network below. The input $X^0 \in \mathbb{R}^{w \times b}$ is a batch of $b$ samples of dimension $w$ that is sampled from a Gaussian distribution with mean $\mathbf{E}(X_{ij}^0) = 0$ and covariance $\mathbf{Cov}(X_{ik}^0, X_{jl}^0) = \delta_{ij}\delta_{kl}$, where $\delta_{ij}$ is the dirac delta function. The first dimension corresponds to the features and the second dimension corresponds to the batch. Let $W^0 \in \mathbb{R}^{w \times w}$ denote the linear layer before the first residual block, and for $\ell > 0$, let $W^\ell$ denote the linear layer on the residual branch of the $\ell$-th residual block (we assume that all layers have the same width $w$ for clarity in presentation). For each weight matrix $W^\ell$, we assume that the elements of $W^\ell$ are independently sampled from $\mathcal{N}(0, 2/w)$ (He initialization). Let $X^\ell \in \mathbb{R}^{w \times b}$ denote the input to the $\ell$-th residual block, let $X^+ = \max(X, 0)$ denote the ReLU non-linearity applied component-wise, and let $\mathcal{B}$ denote the batch normalization operation. Thus, the input to the first residual block is given by $X^1 = W^0 \mathcal{B}(X^0)^+$, and the output of the $\ell$-th residual block is given by $X^{\ell+1} = X^\ell + W^\ell \mathcal{B}(X^\ell)^+$ for $\ell > 0$. We want to analyze the batch normalization statistics for each layer. To this end, we begin by considering the input to the first residual block $X^1$. Note that $X_{ij}^1 = \sum_k W_{ik}^0 \mathcal{B}(X^0)_{kj}^+$. The mean activation $\mathbf{E}(X_{ij}^1) = 0$, while the covariance,

$$
\begin{aligned}
\mathbf{Cov}(X_{ij}^1, X_{lm}^1) &= \mathbf{E}\Big( \sum_{kn} W_{ik}^0 \mathcal{B}(X^0)_{kj}^+ W_{ln}^0 \mathcal{B}(X^0)_{nm}^+ \Big) = \frac{2}{w}\delta_{il} \sum_k \mathbf{E}\Big( \mathcal{B}(X^0)_{kj}^+ \mathcal{B}(X^0)_{km}^+ \Big) \\
&\approx \delta_{il}\left( \frac{1 + (\pi - 1)\delta_{jm}}{\pi} \right).
\end{aligned}
\tag{1}
$$

Since the components of $X^0$ are independent and Gaussian distributed, we have assumed that the components of $\mathcal{B}(X^0)$ are also independent and Gaussian distributed with mean $\mathbf{E}(X_{ij}^0) = 0$ and

$\mathbf{Var}(X_{ij}^0) = 1$. This approximation is tight when the batch size is large ($b \gg 1$). It implies that $\mathbf{E}\big((\mathcal{B}(X^0)_{kj}^+)^2\big) \approx 1/2$, and $\mathbf{E}\big(\mathcal{B}(X^0)_{kj}^+\big) \approx \sqrt{1/2\pi}$, from which we arrive at the equation 1.

We now consider the input to the second residual block $X^2 = X^1 + W^1\mathcal{B}(X^1)^+$. To considerably simplify the analysis, we assume that the width $w$ is large ($w \gg 1$). This implies that $X^1$ is Gaussian distributed with the covariance derived in equation 1 (See [13] for details). Once again, this implies that if the batch size $b$ is also large then the components of $\mathcal{B}(X^1)$ are independent and Gaussian distributed with mean $\mathbf{E}(\mathcal{B}(X^1)_{ij}) = 0$ and $\mathbf{Var}(\mathcal{B}(X^1)_{ij}) = 1$ (note that batch normalization will remove the correlations between different examples in the batch in equation 1). This implies,

$$\mathbf{Cov}((W^1\mathcal{B}(X^1)^+)_{ij}, (W^1\mathcal{B}(X^1)^+)_{lm}) \approx \delta_{il}\left(\frac{1+(\pi-1)\delta_{jm}}{\pi}\right).$$

Furthermore, note that the covariance between the output of the residual branch and the skip connection, $\mathbf{Cov}((W^1\mathcal{B}(X^1)^+)_{ij}, X_{lm}^1) = 0$. We therefore conclude that,

$$\begin{aligned}\mathbf{Cov}(X_{ij}^2, X_{lm}^2) &= \mathbf{Cov}(X_{ij}^1, X_{lm}^1) + \mathbf{Cov}((W^1\mathcal{B}(X^1)^+)_{ij}, (W^1\mathcal{B}(X^1)^+)_{lm}) \\ &\approx 2\delta_{il}\left(\frac{1+(\pi-1)\delta_{jm}}{\pi}\right).\end{aligned}$$

By induction, we can now see that the components of $\mathcal{B}(X^\ell)$ are independent and Gaussian distributed for all $\ell$, and $\mathbf{Cov}((W^\ell\mathcal{B}(X^\ell)^+)_{ij}, X_{lm}^\ell) = 0$ for all $\ell$. Thus, we get,

$$\mathbf{Cov}(X_{ij}^\ell, X_{lm}^\ell) \approx \ell\delta_{il}\left(\frac{1+(\pi-1)\delta_{jm}}{\pi}\right).$$

We are now ready to compute the expected values of the batch statistics, which we denote by $\mu^\ell$ and $\sigma^\ell$ (see appendix A). The expected mean squared activation for a batch of examples on a single channel (the expected squared BatchNorm moving mean),

$$\mathbf{E}\left((\mu_c^\ell)^2\right) = \mathbf{E}\left(\left(\frac{1}{b}\sum_j X_{cj}^\ell\right)^2\right) = \frac{1}{b^2}\sum_{jk}\mathbf{E}\left(X_{cj}^\ell X_{ck}^\ell\right) \approx \ell\left(\frac{1}{\pi} + \frac{\pi-1}{\pi b}\right) \approx \ell/\pi.$$

Meanwhile the expected variance across a batch of examples on a single channel (the expected BatchNorm moving variance),

$$\begin{aligned}\mathbf{E}\left((\sigma_c^\ell)^2\right) &= \mathbf{E}\left(\frac{1}{b}\sum_j (X_{cj}^\ell)^2 - \left(\frac{1}{b}\sum_j X_{cj}^\ell\right)^2\right) \\ &= \mathbf{E}\left(\frac{1}{b}\sum_j X_{cj}^\ell X_{cj}^\ell\right) - \mathbf{E}\left((\mu_c^{\mathcal{B}_\ell})^2\right) \approx \ell(1-1/\pi).\end{aligned}$$

These predictions exactly match our observations in figure 5. Our analysis shows how ReLU non-linearities introduce correlations in the hidden activations between training examples (for shared random weights). These correlations cause the moving variance of the batch normalization layer, which is evaluated on a single channel for a single set of weights, to differ from the variance of the hidden activations over multiple random initializations (which we derived in section 2.1).

# D  Additional results on CIFAR-10

## D.1  Optimal training losses corresponding to table 1

In table 3, we provide the minimum training losses, as well as the optimal learning rates at which the training loss is minimized, when training an $n$-2 Wide-ResNet for a range of depths $n$ on CIFAR-10. At each depth, we train for 200 epochs following the training procedure described in section 2.3 of the main text. These results correspond to the same architectures considered in table 1, where we provided the associated test set accuracies. We provide the training loss excluding the L2 regularization term (i.e., the training set cross entropy), since one cannot meaningfully compare the L2 regularization penalty of normalized and unnormalized networks. These results confirm that batch normalization and SkipInit achieve similar training losses after the same number of training epochs.

Table 3: The training losses, and associated optimal learning rates, of an $n$-2 Wide-ResNet at a range of depths $n$. We train on CIFAR-10 for 200 epochs with either batch normalization or SkipInit.

| Batch Normalization | | | SkipInit ($\alpha = 1$) | | |
|---|---|---|---|---|---|
| **Depth** | **Training loss** | **Learning rate** | **Depth** | **Training loss** | **Learning rate** |
| 16 | $0.007 \pm 0.000$ | $2^{-2}$ ($2^{-2}$ to $2^{-2}$) | 16 | $0.004 \pm 0.000$ | $2^{-3}$ ($2^{-4}$ to $2^{-3}$) |
| 100 | $0.001 \pm 0.000$ | $2^{-3}$ ($2^{-3}$ to $2^{-2}$) | 100 | – | – |
| 1000 | $0.001 \pm 0.000$ | $2^{-3}$ ($2^{-4}$ to $2^{-3}$) | 1000 | – | – |

| SkipInit ($\alpha = 1/\sqrt{d}$) | | | Divide residual block by $\sqrt{2}$ | | |
|---|---|---|---|---|---|
| **Depth** | **Training loss** | **Learning rate** | **Depth** | **Training loss** | **Learning rate** |
| 16 | $0.004 \pm 0.000$ | $2^{-3}$ ($2^{-3}$ to $2^{-3}$) | 16 | $0.013 \pm 0.000$ | $2^{-3}$ ($2^{-3}$ to $2^{-3}$) |
| 100 | $0.001 \pm 0.000$ | $2^{-4}$ ($2^{-4}$ to $2^{-4}$) | 100 | $0.066 \pm 0.015$ | $2^{-6}$ ($2^{-6}$ to $2^{-6}$) |
| 1000 | $0.001 \pm 0.000$ | $2^{-3}$ ($2^{-4}$ to $2^{-3}$) | 1000 | – | – |

| SkipInit ($\alpha = 0$) | | | SkipInit without L2 ($\alpha = 0$) | | |
|---|---|---|---|---|---|
| **Depth** | **Training loss** | **Learning rate** | **Depth** | **Training loss** | **Learning rate** |
| 16 | $0.004 \pm 0.000$ | $2^{-3}$ ($2^{-3}$ to $2^{-3}$) | 16 | $0.008 \pm 0.000$ | $2^{-3}$ ($2^{-3}$ to $2^{-3}$) |
| 100 | $0.001 \pm 0.000$ | $2^{-4}$ ($2^{-4}$ to $2^{-3}$) | 100 | $0.001 \pm 0.000$ | $2^{-3}$ ($2^{-4}$ to $2^{-2}$) |
| 1000 | $0.001 \pm 0.000$ | $2^{-4}$ ($2^{-4}$ to $2^{-4}$) | 1000 | $0.000 \pm 0.000$ | $2^{-2}$ ($2^{-2}$ to $2^{-2}$) |

Table 4: The optimal test accuracies, and associated learning rates, of $n$-2 Wide-ResNets at a range of depths $n$. We train on CIFAR-10 for 200 epochs with different batch-normalized network variants.

| Divide batch-normalized residual block by $\sqrt{2}$ | | |
|---|---|---|
| **Depth** | **Test accuracy** | **Learning rate** |
| 16 | $93.5 \pm 0.2$ | $2^{-2}$ ($2^{-2}$ to $2^{0}$) |
| 100 | $94.6 \pm 0.1$ | $2^{-1}$ ($2^{-1}$ to $2^{-0}$) |
| 1000 | – | – |

| Adding BatchNorm at end of residual block | | |
|---|---|---|
| **Depth** | **Test accuracy** | **Learning rate** |
| 16 | $93.5 \pm 0.1$ | $2^{-1}$ ($2^{-2}$ to $2^{-1}$) |
| 100 | $94.5 \pm 0.1$ | $2^{0}$ ($2^{-1}$ to $2^{0}$) |
| 1000 | – | – |

| Including only the final BatchNorm layer | | |
|---|---|---|
| **Depth** | **Test accuracy** | **Learning rate** |
| 16 | $93.0 \pm 0.1$ | $2^{-1}$ ($2^{-1}$ to $2^{-1}$) |
| 100 | $94.1 \pm 0.1$ | $2^{-2}$ ($2^{-2}$ to $2^{-1}$) |
| 1000 | – | – |

### D.2 Variations of batch-normalized residual networks

To further test the theory that the key benefit of batch normalization in deep residual networks is that it downscales the residual branch at initialization, we now experiment on different variants of batch-normalized residual networks. Using the same training setup as in section 2.3, we show in table 4 that none of the following schemes are able to train a 1000 layer deep Wide-ResNet on CIFAR-10:

- Including batch normalization layers on the residual branch as in a standard Wide-ResNet but multiplying the residual block by $1/\sqrt{2}$ (after the skip path and residual branch merge).

- Including batch normalization layers on the residual branch, as well as including an additional batch normalization layer on the skip path.

- First removing all batch normalization layers, and then placing a single batch normalization layer before the final softmax layer.

|                    |                    |
|:------------------:|:------------------:|
| (a) (for test accuracy) | (b) (for training loss) |

Figure 6: The optimal learning rates of SkipInit, Regularized SkipInit and Batch Normalization, for a 16-4 Wide-ResNet trained for 200 epochs on CIFAR-10. We evaluate the batch statistics over the full training minibatch. All three methods have similar optimal learning rates in the small batch limit.

In all of these experiments, we expect the skip path and the residual branch to contribute equally to the output of the residual block at initialization. Therefore, as predicted by our theory and confirmed by our experiments in table 4, the network becomes harder to train as the depth increases.

### D.3 Optimal learning rates corresponding to figure 4

Finally, in figure 6 we provide the optimal learning rates of SkipInit, Regularized SkipInit and Batch Normalization, when training a 16-4 Wide-ResNet on CIFAR-10. These optimal learning rates correspond to the training losses and test accuracies provided in figure 4 of the main text. The batch statistics for batch normalization layers are evaluated over the full training minibatch.

## E Additional results on CIFAR-100

In tables 5 and 6, we provide the optimal test accuracies and optimal training losses, and the corresponding optimal learning rates, when training $n$-2 WideResNets on CIFAR-100 for different depths $n$ for 200 epochs. We follow the training protocol described in section 2.3 of the main text. Both batch normalization and SkipInit are able to train very deep Wide-ResNets on CIFAR-100.

In figure 7, we compare the performance of SkipInit, Regularized SkipInit (drop probability 0.6), and batch normalization across a wide range of batch sizes, when training a 28-10 Wide-ResNet on CIFAR-100 for 200 epochs. We follow the training protocol described in section 3 of the main text, but we use a ghost batch size of 32. We were not able to train the 28-10 Wide-ResNet to competitive performance when not using either batch normalization or SkipInit. Batch normalized networks achieve higher test accuracies at all batch sizes. However in the small batch limit, the optimal learning rate is proportional to the batch size, and the optimal learning rates of all three methods are approximately equal. As we observed in the main text, batch normalization has a larger maximum stable learning rate, and this allows us to scale training to larger batch sizes.

Finally, in figure 8, we repeat this comparison of SkipInit, Regularized SkipInit and batch normalization at a range of batch sizes, but instead of selecting a fixed ghost batch size, we evauate the batch statistics of batch normalization layers across the full minibatch (as in section 4). We observe a clear regularization effect, whereby the test accuracy achieved with batch normalization peaks for a batch size of 16 and decays rapidly if the batch size is increased or decreased. Regularized SkipInit achieves higher test accuracies than normalized networks when the batch size is small, and it is also competitive with batch normalized networks when the batch size is moderately large. These results emphasize the importance of tuning the ghost batch size in batch normalized networks.

Table 5: The optimal test accuracies and corresponding learning rates (with error bars), when training width 2 Wide-ResNets on CIFAR-100 for a wide range of depths. Both batch normalization and SkipInit are able to train very deep residual networks. However it is not possible to train depth 1000 networks if we do not downscale the hidden activations on the residual branch at initialization.

| | **Batch Normalization** | | | **SkipInit** ($\alpha = 1$) | |
|---|---|---|---|---|---|
| **Depth** | **Test accuracy** | **Learning rate** | **Depth** | **Test accuracy** | **Learning rate** |
| 16 | $72.6 \pm 0.3$ | $2^0$ ($2^{-1}$ to $2^0$) | 16 | $65.6 \pm 0.4$ | $2^{-4}$ ($2^{-4}$ to $2^{-4}$) |
| 100 | $77.2 \pm 0.2$ | $2^0$ ($2^{-1}$ to $2^0$) | 100 | - | - |
| 1000 | $78.0 \pm 0.1$ | $2^0$ ($2^0$ to $2^0$) | 1000 | - | - |

| | **SkipInit** ($\alpha = 1/\sqrt{d}$) | | | **Divide residual block by** $\sqrt{2}$ | |
|---|---|---|---|---|---|
| **Depth** | **Test accuracy** | **Learning rate** | **Depth** | **Test accuracy** | **Learning rate** |
| 16 | $69.3 \pm 0.2$ | $2^{-1}$ ($2^{-2}$ to $2^{-1}$) | 16 | $69.3 \pm 0.2$ | $2^{-2}$ ($2^{-2}$ to $2^{-1}$) |
| 100 | $74.2 \pm 0.1$ | $2^{-1}$ ($2^{-1}$ to $2^{-1}$) | 100 | $60.2 \pm 1.0$ | $2^{-5}$ ($2^{-5}$ to $2^{-5}$) |
| 1000 | $74.7 \pm 0.3$ | $2^{-1}$ ($2^{-1}$ to $2^{-1}$) | 1000 | - | - |

| | **SkipInit** ($\alpha = 0$) | | | **SkipInit without L2** ($\alpha = 0$) | |
|---|---|---|---|---|---|
| **Depth** | **Test accuracy** | **Learning rate** | **Depth** | **Test accuracy** | **Learning rate** |
| 16 | $69.3 \pm 0.2$ | $2^{-1}$ ($2^{-2}$ to $2^{-1}$) | 16 | $63.5 \pm 0.3$ | $2^{-3}$ ($2^{-3}$ to $2^{-3}$) |
| 100 | $74.3 \pm 0.3$ | $2^{-1}$ ($2^{-1}$ to $2^{-1}$) | 100 | $66.0 \pm 0.4$ | $2^{-3}$ ($2^{-3}$ to $2^{-3}$) |
| 1000 | $74.7 \pm 0.3$ | $2^{-1}$ ($2^{-1}$ to $2^{-1}$) | 1000 | $67.9 \pm 0.3$ | $2^{-2}$ ($2^{-2}$ to $2^{-2}$) |

Table 6: The optimal training losses and corresponding learning rates (with error bars), when training width 2 Wide-ResNets on CIFAR-100 for a wide range of depths. Both batch normalization and SkipInit are able to train very deep residual networks. We show it is not possible to train depth 1000 networks if we do not downscale the hidden activations on the residual branch at initialization.

| | **Batch Normalization** | | | **SkipInit** ($\alpha = 1$) | |
|---|---|---|---|---|---|
| **Depth** | **Training loss** | **Learning rate** | **Depth** | **Training loss** | **Learning rate** |
| 16 | $0.078 \pm 0.002$ | $2^{-2}$ ($2^{-2}$ to $2^{-2}$) | 16 | $0.089 \pm 0.022$ | $2^{-4}$ ($2^{-5}$ to $2^{-4}$) |
| 100 | $0.002 \pm 0.000$ | $2^{-2}$ ($2^{-2}$ to $2^{-2}$) | 100 | - | - |
| 1000 | $0.001 \pm 0.000$ | $2^0$ ($2^0$ to $2^0$) | 1000 | - | - |

| | **SkipInit** ($\alpha = 1\sqrt{d}$) | | | **Divide residual block by** $\sqrt{2}$ | |
|---|---|---|---|---|---|
| **Depth** | **Training loss** | **Learning rate** | **Depth** | **Training loss** | **Learning rate** |
| 16 | $0.050 \pm 0.003$ | $2^{-3}$ ($2^{-3}$ to $2^{-3}$) | 16 | $0.062 \pm 0.002$ | $2^{-3}$ ($2^{-3}$ to $2^{-3}$) |
| 100 | $0.002 \pm 0.000$ | $2^{-4}$ ($2^{-4}$ to $2^{-2}$) | 100 | $0.394 \pm 1.270$ | $2^{-5}$ ($2^{-8}$ to $2^{-5}$) |
| 1000 | $0.001 \pm 0.000$ | $2^{-4}$ ($2^{-5}$ to $2^{-2}$) | 1000 | - | - |

| | **SkipInit** ($\alpha = 0$) | | | **SkipInit without L2** ($\alpha = 0$) | |
|---|---|---|---|---|---|
| **Depth** | **Training loss** | **Learning rate** | **Depth** | **Training loss** | **Learning rate** |
| 16 | $0.052 \pm 0.007$ | $2^{-3}$ ($2^{-3}$ to $2^{-2}$) | 16 | $0.122 \pm 0.014$ | $2^{-3}$ ($2^{-4}$ to $2^{-3}$) |
| 100 | $0.002 \pm 0.000$ | $2^{-4}$ ($2^{-4}$ to $2^{-2}$) | 100 | $0.004 \pm 0.000$ | $2^{-3}$ ($2^{-4}$ to $2^{-3}$) |
| 1000 | $0.001 \pm 0.000$ | $2^{-4}$ ($2^{-5}$ to $2^{-3}$) | 1000 | $0.002 \pm 0.000$ | $2^{-3}$ ($2^{-3}$ to $2^{-3}$) |

(a)             (b)

Figure 7: The optimal test accuracy, and the corresponding optimal learning rates of a 28-10 Wide-ResNet, trained on CIFAR-100 for 200 epochs. We were unable to train this network reliably without batch normalization or SkipInit (not shown). Batch normalized networks achieve higher test accuracies, and are also stable at larger learning rates, which enables large batch training.

(a)             (b)

Figure 8: The optimal test accuracy, and the corresponding optimal learning rates of a 28-10 Wide-ResNet, trained on CIFAR-100 for 200 epochs. We do not use ghost batch normalization here, evaluating the batch statistics over the full minibatch. The test accuracy achieved with batch normalization depends strongly on the batch size, and is maximized for a batch size of 16. Regularized SkipInit achieves higher test accuracies than batch normalized networks when the batch size is very small, and it is competitive with batch normalized networks when the batch size is moderately large.

# F    Additional results on ImageNet

In table 7, we present the performance of batch normalization, Fixup and Regularized SkipInit when training Resnet-50-V1 [2] on ImageNet for 90 epochs. Unlike ResNet-V2 and Wide-ResNets, this network does not have an identity skip path, because it introduces a ReLU at the end of the residual block after the skip connection and residual branch merge. We find that Fixup performs slightly worse than batch normalization when the batch size is small, but considerably worse than batch normalization when the batch size is large (similar to the results on ResNet-50-V2). However, Regularized SkipInit is significantly worse than batch normalization and Fixup at all batch sizes. This is not surprising, since we designed SkipInit for models which contain an identity skip connection through the residual block. We also consider a modified version of Regularized SkipInit, which contains a single additional scalar bias in each residual block, just before the final ReLU (after the skip connection and residual branch merge). This scalar bias eliminates the gap in validation accuracy between Fixup and Regularized SkipInit when the batch size is small. We conclude that only two components of Fixup are essential to train the original ResNet-V1: initializing the residual branch at zero, and introducing a scalar bias after the skip connection and residual branch merge.

Table 7: We train ResNet50-V1 on ImageNet for 90 epochs. Fixup performs well when the batch size is small, but performs poorly when the batch size is large. Regularized SkipInit performs poorly at all batch sizes, but its performance improves considerably if we add a scalar bias before the final ReLU in each residual block (after the skip connection and residual branch merge). We perform a grid search to identify the optimal learning rate which maximizes the top-1 validation accuracy. We perform a single run at each learning rate and report both top-1 and top-5 accuracy scores. We use a drop probability of 0.2 for Regularized SkipInit. We note that ResNet-V1 does not have an identity skip connection, which explains why Regularized SkipInit performs poorly without scalar biases.

| | Batch size | | |
| --- | --- | --- | --- |
| **Test accuracy:** | 256 | 1024 | 4096 |
| Batch normalization | 75.6 / 92.5 | 75.3 / 92.4 | 75.4 / 92.4 |
| Fixup | 74.4 / 91.6 | 74.4 / 91.7 | 72.4 / 90.3 |
| Regularized SkipInit | 70.0 / 89.2 | 68.4 / 87.8 | 68.2 / 87.9 |
| Regularized SkipInit + Scalar Bias | 75.2 / 92.4 | 74.9 / 92.0 | 70.8 / 89.6 |

## G    Tensorflow code for ResNet50-V2 with SkipInit

In this section, we provide reference code for our ResNet50-V2 model with Regularized SkipInit using Sonnet [49] in Tensorflow [36].

```python
import collections
import sonnet as snt
import tensorflow as tf

ResNetBlockParams = collections.namedtuple(
    "ResNetBlockParams", ["output_channels", "bottleneck_channels", "stride"])

BLOCKS_50 = (
    (ResNetBlockParams(256, 64, 1),) * 2 + (ResNetBlockParams(256, 64, 2),),
    (ResNetBlockParams(512, 128, 1),) * 3 + (ResNetBlockParams(512, 128, 2),),
    (ResNetBlockParams(1024, 256, 1),) * 5 + (ResNetBlockParams(1024, 256, 2),),
    (ResNetBlockParams(2048, 512, 1),) * 3)

def fixed_padding(inputs, kernel_size, rate=1):
  """Pads the input along spatial dimensions independently of input size."""
  kernel_size_effective = kernel_size + (kernel_size - 1) * (rate - 1)
  pad_total = kernel_size_effective - 1
  pad_begin = pad_total // 2
  pad_end = pad_total - pad_begin
  padded_inputs = tf.pad(inputs, [[0, 0], [pad_begin, pad_end],
                                  [pad_begin, pad_end], [0, 0]])
  return padded_inputs

def _max_pool2d_fixed_padding(inputs,
                              kernel_size,
                              stride,
                              padding,
                              scope=None,
                              **kwargs):
  """Strided 2-D max-pooling with fixed padding (independent of input size)."""
  if padding == "SAME" and stride > 1:
    padding = "VALID"
    inputs = fixed_padding(inputs, kernel_size)
  return tf.contrib.layers.max_pool2d(
      inputs,
```

```
                kernel_size,
                stride=stride,
                padding=padding,
                scope=scope,
                **kwargs)

def _conv2d_same(inputs,
                 num_outputs,
                 kernel_size,
                 stride,
                 rate=1,
                 use_bias=None,
                 initializers=None,
                 regularizers=None,
                 partitioners=None):
  """Strided 2-D convolution with 'SAME' padding."""
  if stride == 1:
    padding = "SAME"
  else:
    padding = "VALID"
    inputs = fixed_padding(inputs, kernel_size, rate)

  return snt.Conv2D(
      num_outputs,
      kernel_size,
      stride=stride,
      rate=rate,
      padding=padding,
      use_bias=use_bias,
      initializers=initializers,
      regularizers=regularizers,
      partitioners=partitioners)(inputs)

class ResNetBlock(snt.AbstractModule):
  """The ResNet subblock, see https://arxiv.org/abs/1512.03385 for details."""

  def __init__(self,
               output_channels,
               bottleneck_channels,
               stride,
               rate=1,
               initializers=None,
               regularizers=None,
               partitioners=None,
               name="resnet_block"):
    """Create a ResNetBlock object for use with the ResNet modules."""
    super(ResNetBlock, self).__init__(name=name)
    self._output_channels = output_channels
    self._bottleneck_channels = bottleneck_channels
    self._stride = stride
    self._rate = rate
    self._initializers = initializers
    self._regularizers = regularizers
    self._partitioners = partitioners

  def _build(self, inputs, is_training=True, test_local_stats=True):
    """Connects the ResNetBlock module into the graph."""
```

```python
num_input_channels = inputs.get_shape()[-1]
with tf.variable_scope("preact"):
  preact = inputs
  preact = tf.nn.relu(preact)

if self._output_channels == num_input_channels:
  if self._stride == 1:
    shortcut = inputs
  else:
    shortcut = _max_pool2d_fixed_padding(
        inputs, 1, stride=self._stride, padding="SAME")
else:
  with tf.variable_scope("shortcut"):
    shortcut = preact
    shortcut = snt.Conv2D(
        self._output_channels, [1, 1],
        stride=self._stride,
        use_bias=True,
        initializers=self._initializers,
        regularizers=self._regularizers,
        partitioners=self._partitioners)(shortcut)

with tf.variable_scope("r1"):
  residual = snt.Conv2D(
      self._bottleneck_channels, [1, 1],
      stride=1,
      use_bias=True,
      initializers=self._initializers,
      regularizers=self._regularizers,
      partitioners=self._partitioners)(preact)
  residual = tf.nn.relu(residual)

with tf.variable_scope("r2"):
  residual = _conv2d_same(
      residual,
      self._bottleneck_channels,
      3,
      self._stride,
      rate=self._rate,
      use_bias=True,
      initializers=self._initializers,
      regularizers=self._regularizers,
      partitioners=self._partitioners)
  residual = tf.nn.relu(residual)

with tf.variable_scope("r3"):
  residual = snt.Conv2D(
      self._output_channels, [1, 1],
      stride=1,
      use_bias=True,
      initializers=self._initializers,
      regularizers=self._regularizers,
      partitioners=self._partitioners)(residual)

# SkipInit
res_multiplier = tf.Variable(0.0, dtype=tf.float32)
residual = res_multiplier*residual
```

```python
    output = shortcut + residual

    return output

def _build_resnet_blocks(inputs,
                         blocks,
                         initializers=None,
                         regularizers=None,
                         partitioners=None):
  """Connects the resnet block into the graph."""
  outputs = []

  for num, subblocks in enumerate(blocks):
    with tf.variable_scope("block_{}".format(num)):
      for i, block in enumerate(subblocks):
        args = {
            "name": "resnet_block_{}".format(i),
            "initializers": initializers,
            "regularizers": regularizers,
            "partitioners": partitioners
        }
        args.update(block._asdict())
        inputs = ResNetBlock(**args)(inputs)
        outputs += [inputs]

  return outputs

class ResNetV2(snt.AbstractModule):
  """ResNet V2 as described in https://arxiv.org/abs/1512.03385."""

  def __init__(self,
               blocks=BLOCKS_50,
               num_classes=1000,
               use_global_pool=True,
               initializers=None,
               regularizers=None,
               partitioners=None,
               custom_getter=None,
               name="resnet_v2"):
    """Creates ResNetV2 Sonnet module."""
    super(ResNetV2, self).__init__(custom_getter=custom_getter, name=name)
    self.blocks = tuple(blocks)
    self._num_classes = num_classes
    self._use_global_pool = use_global_pool
    self._initializers = initializers
    self._regularizers = regularizers
    self._partitioners = partitioners

  def _build(self,
             inputs,
             is_training=True,
             get_intermediate_activations=False):
    """Connects the ResNetV2 module into the graph."""

    outputs = []
    with tf.variable_scope("root"):
```

```python
    inputs = _conv2d_same(
        inputs,
        64,
        7,
        stride=2,
        use_bias=True,
        initializers=self._initializers,
        regularizers=self._regularizers,
        partitioners=self._partitioners)

    inputs = _max_pool2d_fixed_padding(inputs, 3, stride=2, padding="SAME")
    outputs += [inputs]

resnet_outputs = _build_resnet_blocks(
    inputs,
    self.blocks,
    initializers=self._initializers,
    regularizers=self._regularizers,
    partitioners=self._partitioners)

outputs += resnet_outputs

# Take the last layer activations as the input to the next layer.
inputs = resnet_outputs[-1]

with tf.variable_scope("postnorm"):
  inputs = tf.nn.relu(inputs)
  outputs += [inputs]

if self._use_global_pool:
  inputs = tf.reduce_mean(
      inputs, [1, 2], name="use_global_pool", keepdims=True)
  outputs += [inputs]

inputs = tf.contrib.layers.flatten(inputs)

inputs = tf.contrib.layers.dropout(inputs, is_training=is_training,
                                   keep_prob=0.8)

kernel_initializer = tf.contrib.layers.variance_scaling_initializer()

inputs = tf.layers.dense(
    inputs,
    self._num_classes,
    kernel_regularizer=self._regularizers["w"],
    kernel_initializer=kernel_initializer,
    name="logits")

outputs += [inputs]

return outputs if get_intermediate_activations else outputs[-1]
```