[Reviews · NeurIPS 2020]

Review 1

Summary and Contributions: This paper presents an analysis of how batch normalization affects the trainability of residual networks. Based on this analysis, the authors propose an initialization scheme that allows the training of deep unnormalized residual networks.

Strengths: The paper is clearly written and is simple to follow. The main premise of the paper is relevant to the NeurIPS community as residual networks and batch normalization are widely used by practitioners in the field.

Weaknesses: While the theoretical analysis of batchnorm is a very sought after topic, the analysis presented in this paper seems superficial and the actual contribution beyond previous work quite mild. I am specifically bothered with the following : 1. Premise and novelty - The fact that in the forward propagation, batchnorm downscales the variance of the activations in residual branches is hardly surprising, and the suggestion to initialize residual branches to zero for unnormalized networks has been done before in Balduzzi et all. and further expanded upon in Zhang et al. In general, i do not think that analysis of forward propagation statistics is sufficient to analyze the success of back propagation without additional arguments. However, even if we assume that stable forward statistics is all you need to train, the authors have not shed any interesting insight on why batchnorm in particular achieves better performance than unnormalized residual networks. For instance, the "near identity" intialization might not be as important as the authors suggest, since residual networks can also be trained when batch norm is applied only at the output layer. 2.Correctness - Some of the derivations seem dubious, or lack further details. For instance, in section C in the analysis the authors assume that B(x^0) is gaussian if x^0 is gaussian (line 511) which is might be true, but lack any explanation. In addition, the authors intermittently use large width limits to assume Gaussian distributions of pre activations (515). These limits however require more careful arguments when applied together with batchnorm. I do not claim that the claims made in the paper are necessarily false, however further details are missing. In general, i am more bothered by 1.

Correctness: the claims and derivations seem correct, however some details are missing (see my comments above).

Clarity: The paper is clearly written.

Relation to Prior Work: Relation to prior work is poorly discussed. Specifically, i would like the authors to better explain how their work and suggestions differ from those in Balduzzi et all and Zhang et al. Balduzzi et all : Fixup Initialization: Residual Learning Without Normalization (ICLR 2019) Zhang et al.: The Shattered Gradients Problem: If resnets are the answer, then what is the question?

Reproducibility: Yes

Additional Feedback: I would suggest the authors to better position their work with respect to prior work in the field. ***********************************updates********************************** I have read the author’s feedback and other reviewers’ comments The authors partially addressed my concerns with additional experiments. However, i still maintain that the paper is borderline, tending on negative for the following reasons: (1) Significance of results. I feel the main insight of the paper, namely the reason BN networks bias networks towards identity functions is not surprising, and does not offer any non-trivial insights into why BN improve performance. I get that the paper claims it improves trainability, however this seems kind of trivial and obvious since by definition BN normalizes the residual blocks, hence biases towards identity in deep layers. The authors also refute the claim made by other papers that the source of BN success is that it facilitates large learning rate training. I believe this contribution is not significant because the original claim is somewhat void. BN changes the entire loss landscape in nontrivial ways, and so large learning rates cannot by itself explain anything. (2) Practical implications. It seems the results of the experiments do not differ much from Fixup, and still do not match those of BN networks. The simpler to implement claim is also not significant enough since Fixup is already quite easy to implement. Balancing the strengths and weaknesses, I think this paper is borderline, and still tend to be negative. I raise my score to 5. ***********************************updates**********************************


Review 2

Summary and Contributions: This paper makes an interesting observation that the residual blocks become closer to an identity map as the depth of ResNet increases. Motivated by this, the paper proposes a simple initialization scheme SkipInit that can train ResNet without BN. This initialization enlarges the maximum stable learning rate for training, but the generalization performance is worse than ResNet with BN in large-batch training. This challenges previous works which claim that BN regularizes the model by enabling the use of a higher learning rate. This paper also proposes Regularized SkipInit to close the generalization gap between SkipInit and BN.

Strengths: Finding alternatives for BN is an important problem that is interesting to the deep learning community. [17] proposes Fixup initialization, which can be used to initialize normalized ResNet and obtain reasonable performance. This paper can be seen as a followup work of Fixup. Comparing with Fixup, (Regularized) SkipInit is much simpler to implement and can achieve comparable or even better performance than Fixup. Besides, this paper also provides lots of empirical studies on the performances of BN and SkipInit across different batch sizes, which will be a good reference for subsequent works trying to understand the regularization effects of BN.

Weaknesses: The paper challenges the view that BN regularizes the model by enabling the use of a higher learning rate, and regards it as one of the main contributions. But this challenge is actually easy to make. Consider the case that we want to optimize L(w) with initial point w_0 and learning rate eta. For any factor c > 0, we can instead optimize the loss L(w/c) with c*w_0 and c^2*eta, and the training process is exactly the same if we use SGD or other gradient-based methods. Thus, one can enable higher learning rates by simply doing parameterization, and it is meaningless to compare learning rates between different architectures unless there is a suitable normalization for the initialization scale.

Correctness: The theoretical analysis of variance is correct and is verified by the experiments in the paper.

Clarity: This paper is well-written, but Section 2.1 is a bit confusing. It would be better if the authors can organize them into theorems and lemmas in the next version.

Relation to Prior Work: The relationship to related works on initialization is discussed clearly. However, the paper seem to miss many related works on the relationship between BN and large learning rates: if there is no weight decay, SGD with BN can converge with arbitrary learning rates (https://arxiv.org/abs/1906.05890); if there is weight decay, neural networks with BN can be trained with SGD and exponentially increasing learning rates (https://arxiv.org/abs/1910.07454). It is worth to clarify that SkipInit cannot perfectly recover the effects of BN on increasing the maximum stable learning rate.

Reproducibility: Yes

Additional Feedback: - It would be better if the authors could provide more intuitions on why bias + dropout can recover the regularization benefits of batch normalization. In particular, why introducing bias is important here? - In Line 220, the dropout probability is fixed to 0.6 in Regularized SkipInit, but soon it is changed to 0.2 in Table 2. Maybe it should not be fixed to 0.6 at first? ------------------------------ I'm satisfied with the authors' response and now I understand that the authors compared the largest stable LR and the optimal LR within the same architecture. This is an interesting point, so I would like to raise my score to 7.


Review 3

Summary and Contributions: The main finding of this paper is that batch normalization downscales the residual branch relative to a skip connection, by a normalization factor on the order of the square root of the network depth. This in turn ensures the trainability of the network at initialization. The authors develop a simple initialization scheme that can very deep residual networks (1000 layers) without normalization, called SkipInit. The authors also find that the optimal LR for networks w./w.o. batch norm decreases with the batch size, and thus the fact that BN allows larger LR is only helpful for large batch sizes.

Strengths: * This paper is well-structured and very easy to follow. * The main theoretical claim that BN allows training for deep ResNet is because BN downscales the residual branch makes sense. I appreciate the comparison with \sqrt{1/2} initialization, which supports the claim that the right output scale alone is not enough. * The authors also made the interesting empirical observation that under a certain threshold, the optimal learning rate varies linearly w.r.t. the batch size. * The authors give a nice summarization for the three benefits of BN in the discussion section, which are supported by clearly presented experiments.

Weaknesses: * There might be multiple reasons make networks BN trainable under extreme conditions, including large learning rate and huge depth. I agree the point made by this work, that small init in residual branches is such a reason, which in turn makes vanilla resnet withour normalization trainble, however It's possible that the normalized resnet are trainable even without small init in residual branches. It's well known that the input/output scale for the weights before batch normalization is not making as much sense as they do for networks without normalization. For example, Li&Arora, 2019 shows that slightly modified ResNet is trainable with exponential increasing LR and achieves equally good performance as Step Decay schedule. The output of the residual blocks could also grow exponentially, but the network is still trainable because the gradients are small. This phenomemon indeed highlights that the corvariance/scales appeared in the forward pass alone cannot decide trainablity, one has to take back prop into consideration in order to understand what could happen after one step of gradient descent update. This is also the viewpoint hold by Fixup paper. In contrast, this paper doesn't have any arguments for graidents, which makes the explanation less convincing. I would appreciate if the authors could address this issue in the rebuttal. For example, give some intuitive explanation that why introducing a factor of 1/\sqrt{2} would still lead to gradient explosion or something similar. * Continuing the above point, I really wonder what would happen if one introduces the factor of 1/\sqrt{2} to a ResNet with BN. Clearly if the theory of this paper holds, then such networks should not be trainable because the residual branch is not zero comparing to the skip connection. It would also be interesting to see the results for other constants besides 1/\sqrt{2}, such as very small alphas. Note that if the residual link starts with BN, or Conv doesn't have bias, then this trick is essentially downscaling the skip connection alone. When the alpha is very close to 0, the resnet indeed degenerates to a vanilla feedforward CNN. In this special case, the theory in this paper essentially says that even with BN, very deep CNN is not trainable. I would really appreciate if the authors could add the results of above experiments in the future version of the paper, as I think they will be interesting to the community. I will improve my score if the authors could test their theory in the above setting. * The finally proposed methodology is very close to fixup and there's not much improvement over fixup. Also it's always worse than BN. So I would only regard that as a way to justify the theory. ====== Li, Z., & Arora, S. (2019). An exponential learning rate schedule for deep learning. ICLR2020

Correctness: I am not aware of any correctness issue in the theoretical derivation, though I didn't check it very carefully.

Clarity: The paper is well writtern and easy to understand.

Relation to Prior Work: The relation to prior work is clearly discussed.

Reproducibility: Yes

Additional Feedback: ======post-rebuttal update======= I've read the authors' feedback as well as other reviews. I appreciate the authors' extra experimental result in the feedback which convinced me the importance of the init scale. For this reason I changed my score to 7. However, this paper still lacks a direct backward propagation analysis, which limits its theoretical contribution.


Review 4

Summary and Contributions: This paper investigates why batch normalization is important in deep residual networks and shows that batch normalization downscales the hidden activity of the residual branches at initialization, which contributes to the improvement of generalization performance. To demonstrate this, the authors introduce SkipInit, which downscales the residual branch, and confirm that the performance is the same as when using batch normalization. In addition, they confirm experimentally that the availability of large learning rates is not the main benefit of batch normalization.

Strengths: The main contribution of this paper is to clarify both theoretically and experimentally that the downscale of the residual branch at initialization is the most important role played by batch normalization in deep residual networks. In particular, it is a great contribution to show that the proposed SkipInit directly leads to improved generalization performance. The authors have also conducted a large number of experiments on other conventional hypotheses (high learning rates and the effects of regularization) and showed that the benefits they identified in this study were more important than those, which is highly novel information.

Weaknesses: - When introducing \alpha, it is not explained that it is a learnable parameter (only described as "learnable scalar" in Figure 1 and Discussion).

Correctness: - As far as I can see, there is no error in the claims of this paper.

Clarity: This paper is very logical and readers can easily follow the discussion. Experiments are also conducted appropriately according to the development of the discussion.

Relation to Prior Work: The differences with the related work are clearly discussed.

Reproducibility: Yes

Additional Feedback:

[Author Response · NeurIPS 2020]

Before we address the comments raised by each reviewer in turn, we would like to clarify two key points:

**Why normalized ResNets are trainable:** We do not argue that BN can train deep ResNets because it stabilizes the hidden activations on the forward pass. In fact we show in table 1 that stable forward signal propagation is not sufficient (see "divide by $\sqrt{2}$" experiment). Our argument is that, when BN is used, the scale of hidden activations at initialization on the skip path grow proportional to the square root of the depth. The growth of the hidden activations on the skip path is beneficial, because the outputs of normalized residual branches have scale $O(1)$, and therefore the residual blocks in deep BN-ResNets are dominated by the skip path. *Residual blocks dominated by the skip path on the forward pass will also preserve signal propagation on the backward pass.* This is because the backward propagated signal through the $\ell$-th residual branch will be downweighted by $O(\sqrt{\ell})$ relative to the skip path, and therefore the backward signal through the deep normalized residual block is also dominated by the skip path. To provide further evidence for our argument, we have verified empirically that none of the following schemes are able to train a 1000-2 Wide-ResNet:

1. Placing a single BN layer before the softmax (without including BN layers on residual branches).
2. Including BN layers on the residual branch but multiplying the residual block by a factor $\alpha \leq \sqrt{1/2}$.
3. Including BN layers on the residual branch and adding a BN layer after the skip and residual branches merge.

In all of these experiments, the residual branch contributes equally to the output of the residual block at initialization (or it dominates if $\alpha \ll \sqrt{1/2}$). The network thus becomes harder to train as depth increases, as predicted by our analysis.

**Fixup initialization:** The two main contributions of our paper are to explain why deep BN-ResNets are trainable, and to provide a detailed empirical study of the benefits of BN in popular architectures. This empirical study clarifies that large learning rates are not the primary benefit of BN (as claimed by previous papers). The Fixup paper does not study either of these topics, and we therefore believe our paper has a distinct and significant contribution alongside Fixup.

The success of SkipInit also demonstrates that one of the key claims made in the Fixup analysis is false. The authors argue that, in order to train ResNets without BN, one must downscale the weights on the residual branch (even if the hidden activations are already suppressed). Our experiments with SkipInit demonstrate that it is sufficient to downscale the hidden activations. Although SkipInit and Fixup achieve similar performance, SkipInit is simpler to implement.

**Reviewer 1:** Our argument is *not* solely based on the forward pass (see above). Placing a BN layer before the softmax achieves 94.1% test acccuracy for the 100-2 Wide-ResNet on CIFAR-10, but it cannot train the 1000-2 Wide-ResNet.

On line 512 of appendix C, we explain that the approximation $\mathcal{B}(x^0)$ is Gaussian is tight when the batch size $B \gg 1$. Appendix C also assumes large width, and we cite previous work where we build on their results. We do not assume large width in section 2.1. We discuss the work of Balduzzi et al. and Zhang et al. in detail in section 6. See above for further discussion of Zhang et al. (Fixup). We would like to clarify that Balduzzi et al. do not propose that downscaling the residual branch at initialization is sufficient to remove BN. We will expand on these points in the updated text.

**Reviewer 2:** We agree that comparing raw learning rates could be misleading, since one can rescale the learning rate by changing the model parameterization. However this does not affect our argument. Our experiments show that, for small batch sizes, the optimal learning rate with or w/out BN (or with SkipInit) is significantly smaller than the largest stable learning rate. In all cases, the optimal learning rate is only close to the largest stable learning rate when the batch size is large. After changing the model parameterization, the optimal learning rate would still be smaller than the largest stable learning rate for small batch sizes. We therefore conclude that increasing the largest stable learning rate (or improving the model conditioning) is not the main benefit of BN for small batches. We will clarify this point in the updated text.

We will also include a discussion of the suggested references in the updated text, and we will restructure section 2.1 to improve the clarity of presentation. Introducing biases is beneficial because they ensure the expressivity of the model does not fall when BN is removed. Please see the discussion above which clarifies how our work differs from Fixup.

**Reviewer 3:** Please see the summary of our analysis of deep ResNets above. Intuitively, ResNets will preserve signal propagation on both the forward and the backward pass if the residual blocks are dominated by the skip connection. Therefore deep normalized ResNets are trainable since BN downscales the activations on the residual branch. However if we multiply the outputs of residual blocks by $\sqrt{1/2}$, then the residual branch and the skip path will both contribute to the signal on the backward pass. Deep networks of this form will not be trainable with Gaussian weights.

We agree that in principle there could be other benefits of BN which also enable it to train deep networks. However we have tried several variants of BN-ResNets (see discussion above), and every variant we have tried which does not downscale the residual branch at initialization has not been able to train when the depth is large. For example, if we multiply the output of normalized residual blocks by $\alpha = \sqrt{1/2}$, our 100-2 BN-ResNet achieves 94.5% test accuracy on CIFAR-10 but the 1000-2 BN-ResNet does not train. The performance degrades further for coefficients $\alpha < \sqrt{1/2}$.

**Reviewer 4:** Thank you for the positive feedback on our work. We will clarify that each $\alpha$ is a learnable parameter.

[Meta-Review · NeurIPS 2020]

This paper presents an analysis of how batch normalization affects the trainability of residual networks. Based on this analysis, the authors propose an initialization scheme that allows the training of deep unnormalized residual networks. The paper received mixed reviews (clear reject -> marginally below, marginally above -> accept, marginally below -> accept, top 50%). On the positive side, the paper is clearly written, addresses a relevant problem, has solid experiments, overall a very good analysis of the effects of BN. On the negative side, R1 finds that the conclusion not surprising and argues that proposed initialization method had been suggested before by Balduzzi and Zhang. In addition, R1 finds the technical writing lacks clarity to assess correctness. Other reviewers had critiques, but were satisfied by rebuttal and raised their ratings. Overall, I find the strengths outweigh the weaknesses and side with the majority of the reviewers (3/4) that recommend acceptance.